# Association between Immunogenicity of a Monovalent Parenteral P2-VP8 Subunit Rotavirus Vaccine and Fecal Shedding of Rotavirus following Rotarix Challenge during a Randomized, Double-Blind, Placebo-Controlled Trial

**DOI:** 10.3390/v15091809

**Published:** 2023-08-25

**Authors:** Tamika Fellows, Nicola Page, Alan Fix, Jorge Flores, Stanley Cryz, Monica McNeal, Miren Iturriza-Gomara, Michelle J. Groome

**Affiliations:** 1School of Public Health, Faculty of Health Sciences, University of the Witwatersrand, Johannesburg 2001, South Africa; tamikaf@genesis-analytics.com; 2National Institute for Communicable Diseases, A Division of the National Health Laboratory Service, Sandringham 2192, South Africa; nicolap@nicd.ac.za; 3Department of Medical Virology, Faculty of Health Sciences, University of Pretoria, Pretoria 0028, South Africa; 4PATH, Seattle, WA 98121, USAscryz@path.org (S.C.);; 5Department of Pediatrics, University of Cincinnati Medical School, Cincinnati, OH 45229, USA; 6Division of Infectious Diseases, Cincinnati Children’s Hospital Medical Center, Cincinnati, OH 45229, USA; 7South African Medical Research Council Vaccines and Infectious Diseases Analytics Research Unit, Faculty of Health Sciences, University of the Witwatersrand, Johannesburg 2001, South Africa

**Keywords:** rotavirus vaccine, immune response, pediatric, fecal shedding

## Abstract

A correlate of protection for rotavirus (RV) has not been consistently identified. Shedding of RV following an oral rotavirus vaccine (ORV) challenge has been investigated as a potential model to assess protection of parenteral RV vaccines. We previously showed that shedding of a challenge ORV dose was significantly reduced among recipients of a parenteral monovalent RV subunit vaccine (P2-VP8-P[8]) compared to placebo recipients. This secondary data analysis assessed the association between fecal shedding of RV, as determined by ELISA one week after receipt of a Rotarix challenge dose at 18 weeks of age, and serum RV-specific antibody responses, one and six months after vaccination with the third dose of the P2-VP8-P[8] vaccine or placebo. We did not find any association between serum RV-specific immune responses measured one month post-P2-VP8-P[8] vaccination and fecal shedding of RV post-challenge. At nine months of age, six months after the third P2-VP8-P[8] or placebo injection and having received three doses of Rotarix, infants shedding RV demonstrated higher immune responses than non-shedders, showing that RV shedding is reflective of vaccine response following ORV. Further evaluation is needed in a larger sample before fecal shedding of an ORV challenge can be used as a measure of field efficacy in RV vaccine trials.

## 1. Introduction

Rotavirus (RV) occurs globally and causes severe dehydrating diarrheal disease in young children. In 2013, it was estimated that 215,000 children died as a result of RV infection. This corresponded to 37% of all deaths as a result of dehydrating diarrhea amongst children less than 5 years of age, of which more than 90% occurred in low- and middle-income countries (LMIC) [1]. Several oral live-attenuated replicating RV vaccines (ORV) have been World Health Organization (WHO) prequalified and introduced into routine infant vaccination programs in over 100 countries worldwide, leading to significant reductions in disease morbidity and mortality [2,3]. However, these vaccines have reduced efficacy and effectiveness in LMIC compared to high-income countries (HIC) [4,5]. It is thus necessary to develop alternative RV vaccines to further prevent disease in LMIC where the burden of diarrheal disease remains highest.

One of the major factors hindering the development of RV vaccines is the lack of a reliable correlate of protection for RV, which can serve as a proxy indicator of protection against disease [6]. It is not always ethical or financially feasible to assess clinical endpoints through large efficacy trials, considering that oral RV vaccines are available in many countries and the sample size required to achieve sufficient power for noninferiority trials is very large. Serum RV-specific IgA antibody titers have been assessed in a number of vaccine trials evaluating efficacy of ORV and were found to be an important predictor of the immune response protecting children from rotavirus diarrhea [7]. Yet it is unclear whether these antibody measures would adequately predict protection against novel parenteral rotavirus vaccines.

Upon infection with either naturally occurring RV or live-attenuated ORV strains, the viral strains replicate within the gut and are shed in the stool until the immune system can mount sufficient response to clear the infection [8,9,10]. The presence of fecal vaccine shedding would, therefore, suggest lack of intestinal mucosal immunity sufficient to clear infection, and fecal shedding following an ORV “challenge” dose could potentially be used as a proxy measure of efficacy. The absence of fecal shedding following a second dose of Rotarix in Bangladeshi infants was associated with a decreased risk of RV diarrhea in the first two years of life [11]. Clinical trials evaluating monovalent and trivalent formulations of a novel nonreplicating parenteral rotavirus vaccine, a P2-VP8 subunit, assessed fecal shedding of rotavirus following administration of an ORV (Rotarix) as a potential proxy for RV vaccine efficacy [12,13]. There was significant reduction in shedding in some of the vaccine dose groups compared to the placebo recipients, suggesting local suppression of the oral vaccine strain following parenteral vaccination.

This secondary data analysis assessed the association between fecal shedding of RV following a challenge dose of Rotarix and serum RV-specific antibody responses, one and six months after vaccination with the third dose of the parenteral monovalent RV subunit vaccine (P2-VP8-P[8]) or placebo.

## 2. Materials and Methods

The primary study was a dose-escalation, randomized, placebo-controlled trial to investigate the safety and immunogenicity of the P2-VP8-P[8] vaccine in healthy, HIV-uninfected toddlers and infants conducted at the Respiratory and Meningeal Pathogens Research Unit, Johannesburg, South Africa during 2014–2015. The infant cohort included term infants (gestation period > 37 weeks) aged from 6 to <8 weeks at time of enrolment and determined to be in good health through medical history, physical examination, and laboratory testing. A parent or legal guardian provided written informed consent for their child’s participation in the trial. In total, 162 infants were enrolled and randomized to receive vaccine (*n* = 12 in 10 µg group, *n* = 50 in 30 µg group and *n* = 50 in 60 µg group) or placebo (*n* = 50). Infants received three doses of P2-VP8-P[8] vaccine or placebo at approximately 6, 10 and 14 weeks of age. Serum samples were collected at baseline (Day 0: 6 to <8 weeks of age) and one and six months after receiving the third dose of the P2-VP8-P[8] vaccine or placebo (Day 84: ~18 weeks of age; and Day 224: ~9 months of age, respectively; Appendix A).

All immunogenicity testing was performed as part of the original study. In brief, IgA and IgG titers directed against P2-VP8-P[8] vaccine antigen (anti-P[8] IgA and IgG) and the whole rotavirus lysate (anti-RV IgA) were measured, as well as neutralizing antibody (NAb) responses against four RV strains: Wa (G1P[8]), 89-12 (G1P[8]), DS-1 (G2P[4]), and 1076 (G2P[6]), as described [14]. As low NAb responses to heterologous P[4] and P[6] strains were previously demonstrated [12], and responses against DS-1 and 1076 strains were not measured at the six-month post-vaccination time point, only NAb against Wa and 89-12 were assessed in the current analysis. Placebo and vaccine groups received three doses of Rotarix (one month apart starting at 18 weeks of age) after receipt of the P2-VP8-P[8] vaccine/placebo. Fecal shedding of rotavirus was measured 5, 7 and 9 days after the first Rotarix dose by ELISA using the ProsPect™ Rotavirus Microplast Assay (Oxoid Ltd., Ely, United Kingdom), according to the manufacturer’s instructions. Additional details on the study design and randomization have been described previously [12].

Infants who had received three injections of either of the two highest vaccine doses (30 µg or 60 µg) or placebo, had a serum sample collected four weeks after the last injection and provided at least one stool sample following administration of the first dose of Rotarix were included in the analysis. Infants with at least one stool sample testing rotavirus positive by ELISA were considered to be shedders and those with all stool samples testing ELISA negative were considered to be non-shedders. IgA, IgG, and NAb were analyzed in terms of both seroresponse and magnitude of the response. Unadjusted seroresponse was defined as a four-fold or more increase in the magnitude of the antibody response from baseline to one or six months after receiving three doses of P2-VP8-P[8] vaccine or placebo. The magnitude of the antibody response was described by geometric mean titers (GMT). IgG and NAb titers at the one month post-third-dose time point were adjusted for decay in maternal antibodies using the half-life calculated from participants in the placebo group who had detectable baseline titers that were higher than at the post-injection visit This was established separately for each assay. Adjusted seroresponse was defined as a four-fold or more increase in titer from baseline to one month after receiving three doses of P2-VP8-P[8] vaccine or placebo (adjusted titer) in infants with an unadjusted post-injection titer greater than the limit of detection [12]. Exposure to HIV was determined from history from the mother and use of nevirapine prophylaxis in the infant.

Continuous variables were assessed for normality and means or geometric means with 95% confidence intervals (CI) described, as appropriate. Wilcoxon Rank-Sum test or Student’s *t*-test for continuous variables, and Pearson’s Chi-Squared or Fishers Exact (where cells values were 5 or less) tests for categorical variables were conducted to determine if there were differences in the distribution of demographic and immune response factor variables between those shedding and those not shedding RV. Separate comparisons were performed for P2-VP8-P[8] vaccine (30 µg and 60 µg combined) and placebo recipients. *p*-values < 0.05 were considered significant. STATA version 16.1 (StataCorp, College Station, TX, USA) was used for all analyses.

## 3. Results

### 3.1. Association between Vaccination Status and Non-Immunological Factors and Fecal Shedding of Rotavirus

There were 135 infants who received 30 µg P2-VP8-P[8] (*n* = 45), 60 µg P2-VP8-P[8] (*n* = 46) or placebo (*n* = 44), had a serum sample collected four weeks after the last injection and provided at least one stool sample after the first Rotarix dose. Of these, 32 (23.7%) infants shed RV in the stool and 103 (76.3%) did not shed RV. Shedding was present in 16.5% (15/91) of P2-VP8-P[8] vaccine recipients (30 µg and 60 µg doses), which was significantly lower than shedding in placebo recipients (17/44, 38.6%; *p* = 0.005). There were no significant differences in sex, age of infants at enrolment or HIV-exposure status at baseline between shedders and non-shedders among either P2-VP8-P[8] vaccine or placebo recipients (*p* > 0.05 for all comparisons; Appendix A).

### 3.2. Association between Immune Responses and Fecal Shedding of Rotavirus among P2-VP8-P[8] Recipients

One month after the third P2-VP8-P[8] vaccine dose, anti-P2-VP8-P[8] IgA seroresponse was higher in shedders (93.3%) compared to non-shedders (69.3%), although this did not reach statistical significance (*p* = 0.062; Table 1). There were no significant differences in anti-RV IgA, anti-P2-VP8-P[8] IgG or NAb seroresponses to Wa or 89-12 strains between shedders and non-shedders who received 30 µg and 60 µg P2-VP8-P[8] vaccine. Six months after the third P2-VP8-P[8] vaccine dose, and having also received three doses of Rotarix, anti-P2-VP8-P[8] IgA, anti-RV IgA and anti-P2-VP8-P[8] IgG seroresponses were higher in shedders compared to non-shedders, although this did not reach statistical significance for any comparisons. Similarly, NAb seroresponses to Wa and 89-12 strains tended to be higher among shedders compared to non-shedders, although not significantly (Table 1).

There were no significant differences between anti-P2-VP8-P[8] IgA, anti-RV IgA or anti-P2-VP8-P[8] IgG antibody titers between vaccinated shedders and non-shedders at baseline or one month after receiving the third P2-VP8-P[8] vaccine (Table 2). However, at six months post-third P2-VP8-P[8] vaccine, and having received three doses of Rotarix, RV shedders had significantly higher anti-P2-VP8-P[8] IgA titers than non-shedders (*p* = 0.005; Table 2). Anti-RV IgA and anti-P2-VP8-P[8] IgG antibody titers were higher in shedders vs. non-shedders but did not reach statistical significance (*p* = 0.074 and *p* = 0.067, respectively). Interestingly, NAb titers to Wa and 89-12 RV strains were lower in shedders compared to non-shedders at baseline (GMT 64.3 vs. 113.0; *p* = 0.057 and GMT 68.8 vs. 147.5; *p* = 0.018, respectively), did not differ significantly at one month following the third P2-VP8-P[8] dose, and were higher in shedders compared to non-shedders at six months post-third P2-VP8-P[8] dose (234.1 vs. GMT 114.3; *p* = 0.033 and GMT 355.6 vs. 165.9; *p* = 0.015, respectively; Table 2).

### 3.3. Association between Immune Seroresponses and Fecal Shedding of Rotavirus among Placebo Recipients

Anti-P2-VP8-P[8] IgA, anti-RV IgA, and anti-P2-VP8-P[8] IgG seroresponses as well as NAb seroresponses to Wa and 89-12 strains among placebo recipients were relatively low at the Day 84 timepoint and there were no significant differences in seroresponses between shedders and non-shedders (Table 3). At the Day 224 timepoint, following three doses of Rotarix, there were no significant differences in anti-P2-VP8-P[8] IgA and anti-P2-VP8-P[8] IgG seroresponses between shedders and non-shedders (82.4% vs. 70.4%, *p* = 0.486 and 52.9 vs. 29.6, *p* = 0.122, respectively), although responses tended to be higher in shedders compared to non-shedders. Anti-RV IgA seroresponse was significantly higher in shedders (87.5%) compared to non-shedders (48.2%, *p* = 0.021). NAb seroresponses to Wa and 89-12 strains were higher among shedders (41.2% for both strains) compared to non-shedders (11.1% and 18.5%, respectively), although this only reached statistical significance for the Wa strain (*p* = 0.030, Table 3).

Among placebo recipients, there were no significant differences in anti-P2-VP8-P[8] IgA, anti-RV IgA, anti-P2-VP8-P[8] IgG or NAb titers between shedders and non-shedders at baseline or the Day 84 time point (Table 4). However, at the Day 224 time point, and having received three doses of Rotarix, RV shedders had significantly higher anti-P2-VP8-P[8] IgA, anti-RV IgA and anti-P2-VP8-P[8] IgG antibody titers than non-shedders (*p* = 0.001, *p* = 0.004 and *p* = 0.002, respectively; Table 4). NAb titers to Wa and 89-12 RV strains were also significantly higher in shedders compared to non-shedders (193.5 vs. 35.0, *p* < 0.001 and 310.1 vs. 60.5, *p* = 0.001, respectively; Table 4) at the Day 224 time point.

## 4. Discussion

Correlates of protection, which can act as a proxy measure of protection against clinical disease, can be used to enable noninferiority trials to support licensure of vaccines where placebo-controlled trials are no longer ethical or feasible due to low attack rates, to bridge the transition from first-generation to second-generation vaccines, to monitor the consistency of vaccine production, to study the susceptibility of a population after vaccination and to enable the licensure of combination vaccines [15]. For enteric vaccines, correlates of protection are currently associated with serological immune responses even though these may not be indicative of gut immune responses or protection [15]. Several correlates of protection have been suggested for RV but none of these have been consistently understood or validated. Potential correlates of protection for live oral RV vaccines include serum IgA antibody detected by whole RV lysate ELISA, serum NAbs to RV strains, RV-specific serum IgG antibodies, and shedding of RV in stools following vaccination [16,17]. The data analyzed in this publication were from a Phase I/II study that primarily assessed safety and immunogenicity of the monovalent P2-VP8-P[8] subunit vaccine, but also included an assessment of RV fecal shedding following a “challenge” dose of the ORV Rotarix given one month after the third injection of P2-VP8-P[8] vaccine/placebo. RV shedding was found to be significantly lower in infants who were vaccinated with 30 μg and 60 μg doses of P2-VP8-P[8] compared to placebo recipients, suggesting that the vaccine might induce immune responses that can neutralize infection in the gut [12]. As both RV fecal shedding and serum RV-specific immune responses have been proposed as potential correlates of protection for RV, this secondary data analysis evaluated associations between fecal shedding after Rotarix challenge and serum immune responses, observed one and six months after three doses of P2-VP8-P[8] vaccine or placebo. Overall, there was no significant difference in serum immune responses one month after receiving three doses of P2-VP8-P[8] protein subunit vaccine between those shedding RV post-challenge and those not shedding RV.

Anti-RV IgA seroresponse to whole RV lysate was low at the Day 84 time point (one month post-third P2-VP8-P[8] dose or placebo) among non-shedders in both vaccine (8.1%) and placebo (7.4%) recipients. None of the vaccine or placebo recipients who demonstrated RV shedding post-challenge showed any anti-RV IgA seroresponse at the Day 84 time point. This was expected as infants were unlikely to be have been naturally exposed to RV prior to this time point. In addition, in the vaccine group, the parenteral vaccine was unlikely to generate a robust response to the whole RV lysate. However, at the Day 224 time point, when all participants had also received three doses of Rotarix, anti-RV IgA seroresponse and GMTs were higher overall compared to Day 84, and significantly higher in placebo recipients shedding RV after the first Rotarix (challenge dose) compared to non-shedding placebo recipients. This suggests that those shedding RV after the first Rotarix dose demonstrated a robust anti-RV IgA response at around nine months of age. This provides further support to the potential to use shedding following the first dose of ORV as a predictor of vaccine response, with the added benefit of obtaining results earlier and with a noninvasive sample [18].

One month after the third P2-VP8-P[8] dose in vaccine recipients, anti-P2-VP8-P[8] IgA seroresponse and titers were similar in RV shedders compared to non-shedders, suggesting no clear association between RV shedding and anti-P2-VP8-P[8] IgA as measures of immune response to a parenteral RV vaccine. At the Day 224 time point, after receipt of three doses of Rotarix, there were good anti-P2-VP8-P[8] IgA seroresponses in both shedders and non-shedders, although GMTs were significantly higher in shedders compared to non-shedders. This once again suggests that infants who shed RV following their first dose of ORV showed slightly better immune response measured at nine months of age.

Several studies have shown that serum IgG was associated with protection against RV infection and illness [19,20,21]. Serum anti-P2-VP8-P[8] IgG seroresponse one month after the third P2-VP8-P[8] dose approached 100% in both shedders and non-shedders, with slightly higher GMTs in shedders compared to non-shedders. Thus, serum anti-P2-VP8-P[8] IgG was also not clearly associated with fecal shedding after an ORV challenge in vaccinated infants. Following three doses of Rotarix, anti-P2-VP8-P[8] IgG titers were higher in shedders compared to non-shedders in both vaccine and placebo recipients; although only significantly in placebo recipients. Overall, P2-VP8-P[8] vaccine recipients had better immune response and higher titers compared to placebo recipients suggesting that IgG may be a better measure of immune response to a parenteral RV vaccine [12]. There were no significant differences in NAb titers or response to the Wa and 89-12 RV strains measured one month after receipt of the third P2-VP8-P[8] dose among shedders and non-shedders. This suggests that there was also no association between fecal shedding of RV and NAb responses at this time point. At the Day 224 time point, after receipt of three doses of Rotarix, NAb titers to Wa and 89-12 were significantly higher in shedders compared to non-shedders suggesting that NAb are potentially a measure of immune response to an ORV, rather than a parenteral vaccine. The correlation between serum NAb response to different RV strains and protection against RV infection is context and strain specific [6].

At the Day 224 time point, six months after the third P2-VP8-P[8] or placebo injection and having received three doses of Rotarix, RV shedders demonstrated higher immune responses than non-shedders. Those with higher immune responses to IgA anti-P[8], IgA anti-whole RV lysate, NAb to RV strain Wa and 89-12 at around nine months of age were more likely to have shed RV following the first dose of Rotarix (challenge). This shows that RV shedding after ORV challenge is reflective of vaccine response following ORV and is predictive of higher serum immune responses at this later time point. This finding requires further investigation to properly understand what this means for RV shedding as a potential correlate of protection. Infants in both the vaccine and placebo groups would all have received three doses of the ORV, Rotarix, by this time point. The relationships seen between RV shedding and serum immune responses at nine months of age may also possibly be influenced by the same factors that have been hypothesized to be responsible for the lower efficacy and immunogenicity observed in the clinical trials assessing ORVs [22]. Factors such as enteric coinfections at the time of ORV vaccination, breast milk antibodies and host genetic factors such as histo-blood group antigens may affect RV shedding after the challenge dose, and thus confound the relationship between shedding and serum immune responses.

This analysis had several limitations. First, the study was powered for safety and immunogenicity endpoints for the primary study, and the lack of significance may have been due to the sample size available for this analysis. The separate analyses for vaccine recipients and placebos further reduced the power. A larger sample size is needed to further evaluate differences in serum immune responses by shedding status. Important confounders such as Lewis and secretor status, enteric coinfections and breastfeeding were not measured in the primary study and could not be adjusted for in the analysis. Rotavirus-specific copro-IgA and IgG antibodies were not measured, so their association with fecal RV shedding could not be assessed. Rotavirus shedding was due to an attenuated stain, not natural infection with a virulent virus, making it difficult to correlate the findings with protection against disease. Exposure to wild-type RV during the study may also have confounded our results and complicated the interpretation. The strain was confirmed as the Rotarix^®^ vaccine strain in 29/32 (91%) of shedders, with the other three (one vaccine recipient and two placebo recipients) being a G9P[8] strain, which was predominant in the 2015 rotavirus season in South Africa, indicating that they were exposed to a natural infection [12]. As the use of ELISA to detect shedding was being assessed in this study, we did not exclude these participants from the analysis. The majority were shedding the vaccine strain, so this would have had a minimal effect on the results.

## 5. Conclusions

Whilst we previously showed that shedding of challenge ORV dose was significantly reduced among P2-VP8 vaccine recipients compared to placebo, we did not find any association between serum RV-specific immune responses measured one month post-P2-VP8 vaccination and fecal shedding of RV post-challenge. Lack of significance observed could be a result of a limited sample size and additional investigations with increased sample size are required. However, these data may also suggest that the identification of a serological correlate of protection for an injectable subunit rotavirus vaccine may require greater dissection of the immune responses elicited by vaccination. This may include further investigation of the response by IgG subclass, and evaluating cellular Fc effector functions, for example antibody (Ab)-dependent cellular phagocytosis, activated when the Ab Fc region forms immune complexes with antigens and Fc receptors on innate immune cells. In the absence of an absolute immunological correlate of protection for RV, it is difficult to establish the usefulness of this fecal shedding model using only immunological correlations. In order to clearly establish whether shedding post-ORV challenge can be used as a marker of field efficacy, further studies are needed to investigate whether fecal shedding post-ORV challenge is correlated with clinical protection. In addition, any evaluation of shedding needs to take other potential confounders into account, including Lewis and secretor status, breastmilk antibodies, and presence of enteric coinfections at the time of vaccination.

## Figures and Tables

**Table 1 viruses-15-01809-t001:** Serum anti-P2-VP8-P[8] IgA, anti-RV IgA (whole lysate), anti-P2-VP8-P[8] IgG, and neutralizing antibody responses in P2-VP8-P[8] recipients, stratified by rotavirus shedding status (as determined by ELISA one week after receipt of a Rotarix challenge dose at 18 weeks of age).

	30 µg and 60 µg P2-VP8-P[8] Recipients *^a^*					
Seroresponse	Day 84 (One Month Post-Dose 3)			Day 224 (Six Months Post-Dose 3)		
	Not Shedding *^b^*	Shedding *^c^*	*p*-Value	Not Shedding *^b^*	Shedding *^c^*	*p*-Value
Anti-P2-VP8-P[8] IgA	52/75 (69.3)	14/15 (93.3)	0.062 *^d^*	53/72 (73.6)	13/14 (92.9)	0.172 *^d^*
Anti-RV IgA	6/74 (8.1)	0/14 (0)	0.584 *^d^*	53/71 (74.7)	12/13 (92.3)	0.280 *^d^*
Anti-P2-VP8-P[8] IgG	74/76 (97.4)	15/15 (100)	1.000 *^d^*	58/73 (79.5)	13/14 (92.9)	0.451 *^d^*
Anti-P2-VP8-P[8] IgG—adjusted *^e^*	75/76 (98.7)	15/15 (100)	1.000 *^d^*	-	-	-
NAb to RV strain Wa	18/76 (23.7)	5/15 (33.3)	0.517 *^d^*	12/73 (16.4)	5/14 (35.7)	0.137 *^d^*
NAb to RV strain Wa—adjusted *^e^*	63/76 (82.9)	15/15 (100)	0.116 *^d^*	-	-	-
NAb to RV strain 89-12	29/76 (38.2)	7/15 (46.7)	0.538	14/73 (19.2)	6/14 (42.9)	0.054
NAb to RV strain 89-12—adjusted *^e^*	63/76 (82.9)	15/15 (100)	0.116 *^d^*	-	-	-

*^a^* All vaccine recipients received three doses of Rotarix at ~18, 22 and 24 weeks of age. *^b^* ELISA negative, *^c^* ELISA positive, *^d^* Fisher’s exact, *^e^* Adjusted for maternal antibodies.

**Table 2 viruses-15-01809-t002:** Serum anti-P2-VP8-P[8] IgA, anti-RV IgA (whole lysate), anti-P2-VP8-P[8] IgG and neutralizing antibody geometric mean titers in P2-VP8-P[8] recipients, stratified by rotavirus shedding status (as determined by ELISA one week after receipt of a Rotarix challenge dose at 18 weeks of age).

	30 µg and 60 µg P2-VP8-P[8] Recipients *^a^*		
	Not Shedding *^b^*	Shedding *^c^*	*p*-Value *^e^*
**Anti-P2-VP8-P[8] IgA** (GMT (95% CI))			
Day 0	5.8 (5.1–6.7)	5.0 (4.0–6.1)	0.325
Day 84	42.6 (32.2–56.2)	63.1 (36.2–109.9)	0.244
Day 224	61.9 (45.5–84.4)	192.9 (78.2–475.9)	**0.005**
**Anti-RV IgA** (GMT (95% CI))			
Day 0	9.3 (8.2–10.6)	7.5 (7.5–7.5)	0.143
Day 84	11.1 (8.7–14.0)	8.0 (7.0–9.1)	0.228
Day 224	89.0 (65.0–121.7)	179.0 (88.9–353.4)	0.074
**Anti-P2-VP8-P[8] IgG** (GMT (95% CI))			
Day 0	142.5 (104.4–194.5)	101.3 (54.0–190.0)	0.366
Day 84	9074.9 (7695.9–10,701.0)	12,115.8 (9181.2–15,988.4)	0.143
Day 84—adjusted *^d^*	38671.9 (32,711.9–45,717.9)	50,498.6 (38,443.6–66,333.8)	0.1808
Day 224	1780.5 (1458.3–2174.0)	2814.3 (1780.1–4449.5)	0.067
**NAb to RV strain Wa** (GMT (95% CI))			
Day 0	113.0 (89.2–143.0)	64.3 (36.1–114.5)	0.057
Day 84	198.5 (166.6–236.6)	233.1 (182.9–297.1)	0.436
Day 84—adjusted *^d^*	1452.7 (1216.2–1735.2)	1655.9 (1300.4–2108.4)	0.530
Day 224	114.3 (87.2–150.0)	234.1 (133.7–410.0)	**0.033**
**NAb to RV strain 89-12** (GMT (95% CI))			
Day 0	147.5 (114.4–190.3)	68.8 (37.2–127.5)	**0.018**
Day 84	344.9 (283.6–419.5)	431.9 (311.4–599.0)	0.335
Day 84—adjusted *^d^*	2209.4 (1812.2–2693.7)	2691.1 (1968.8–2678.4)	0.401
Day 224	165.9 (129.7–212.2)	355.6 (198.0–638.7)	**0.015**

*^a^* All vaccine recipients received three doses of Rotarix at ~18, 22 and 24 weeks of age *^b^* ELISA negative, *^c^* ELISA positive, *^d^* Adjusted for maternal antibodies, *^e^* Student’s *t*-test on log-transformed data.

**Table 3 viruses-15-01809-t003:** Serum anti-P2-VP8-P[8] IgA, anti-RV IgA (whole lysate), anti-P2-VP8-P[8] IgG and neutralizing antibody responses in placebo recipients, stratified by rotavirus shedding status (as determined by ELISA one week after receipt of a Rotarix challenge dose at 18 weeks of age).

	Placebo Recipients *^a^*					
Seroresponse	Day 84			Day 224		
	Not Shedding *^b^*	Shedding *^c^*	*p*-Value	Not Shedding *^b^*	Shedding *^c^*	*p*-Value
Anti-P2-VP8-P[8] IgA	8/27 (29.6)	1/17 (5.9)	0.121 *^d^*	19/27 (70.4)	14/17 (82.4)	0.486 *^d^*
Anti-RV IgA	2/27 (7.4)	0/16 (0)	0.522 *^d^*	13/27 (48.2)	14/16 (87.5)	**0.021** *^d^*
Anti-P2-VP8-P[8] IgG	1/27 (3.7)	0/17 (0)	1.000 *^d^*	8/27 (29.6)	9/17 (52.9)	0.122
Anti-P2-VP8-P[8] IgG—adjusted *^e^*	3/27 (11.1)	1/17 (5.9)	1.000 *^d^*	-	-	-
NAb to RV strain Wa	0/27 (0)	0/17 (0)	-	3/27 (11.1)	7/17 (41.2)	**0.030** *^d^*
NAb to RV strain Wa -adjusted *^e^*	3/27 (11.1)	0/17 (0)	0.272 *^d^*	-	-	-
NAb to RV strain 89-12	1/27 (3.7)	0 /17 (0)	1.000 *^d^*	5/27 (18.5)	7/17 (41.2)	0.164 *^d^*
NAb to RV strain 89-12—adjusted *^e^*	4/27 (14.8)	0/17 (0)	0.147 *^d^*	-	-	-

*^a^* All placebo recipients received three doses of Rotarix at ~18, 22 and 24 weeks of age *^b^* ELISA negative, *^c^* ELISA positive, *^d^* Fisher’s exact, *^e^* Adjusted for maternal antibodies.

**Table 4 viruses-15-01809-t004:** Serum anti-P2-VP8-P[8] IgA, anti-RV IgA (whole lysate), anti-P2-VP8-P[8] IgG and neutralizing antibody geometric mean titers in placebo recipients, stratified by rotavirus shedding status (as determined by ELISA one week after receipt of a Rotarix challenge dose at 18 weeks of age).

	Placebo Recipients *^a^*		
	Not Shedding *^b^*	Shedding *^c^*	*p*-Value *^e^*
**Anti-P2-VP8-P[8] IgA** (GMT (95% CI))			
Day 0	5.6 (4.5–7.0)	8.3 (5.0–14.0)	0.098
Day 84	13.5 (7.9–23.0)	10.0 (5.9–17.2)	0.453
Day 224	44.6 (24.5–81.1)	289.0 (103.9–804.1)	**0.001**
**Anti-RV IgA** (GMT (95% CI))			
Day 0	10.5 (7.0–15.9)	9.4 (6.9–13.0)	0.701
Day 84	13.3 (7.6–23.2)	7.7 (7.4–7.9)	0.118
Day 224	64.5 (34.1–122.0)	283.2 (129.8–617.6)	**0.004**
**Anti-P2-VP8-P[8] IgG** (GMT (95% CI))			
Day 0	109.3 (57.1–209.2)	76.4 (35.9–162.6)	0.468
Day 84	39.2 (22.6–68.0)	21.9 (12.6–38.0)	0.150
Day 84—adjusted *^d^*	162.6 (92.8–284.9)	93.4 (53.9–161.9)	0.175
Day 224	99.1 (53.5–183.6)	491.4 (217.3–1111.2)	**0.002**
**NAb to RV strain Wa** (GMT (95% CI))			
Day 0	78.8 (47.4–131.0)	74.5 (42.3–131.3)	0.882
Day 84	15.6 (9.5–25.5)	9.8 (5.7–16.8)	0.208
Day 84—adjusted *^d^*	104.5 (61.8–176.9)	71.7 (41.7–123.3)	0.329
Day 224	35.0 (20.3–60.5)	193.5 (90.5–413.6)	**<0.001**
**NAb to RV strain 89-12** (GMT (95% CI))			
Day 0	96.1 (56.9–162.1)	96.9 (47.0–199.6)	0.984
Day 84	25.5 (14.7–44.0)	13.8 (7.9–24.4)	0.132
Day 84—adjusted *^d^*	150.7 (86.4–263.0)	89.1 (50.7–156.4)	0.197
Day 224	60.5 (33.1–110.8)	310.1 (144.3–666.6)	**0.001**

*^a^* All placebo recipients received three doses of Rotarix at ~18, 22 and 24 weeks of age *^b^* ELISA negative, *^c^* ELISA positive, *^d^* Adjusted for maternal antibodies, *^e^* Student’s *t*-test on log-transformed data.

## Data Availability

The data presented in this study are available on request from the corresponding author. The data are not publicly available due to a clinical trial agreement with PATH Vaccine Solutions.

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
