# Peer review of "Association between Immunogenicity of a Monovalent Parenteral P2-VP8 Subunit Rotavirus Vaccine and Fecal Shedding of Rotavirus following Rotarix Challenge during a Randomized, Double-Blind, Placebo-Controlled Trial"

_viruses, 2023, doi:10.3390/v15091809_

Round 1

Reviewer 1 Report

This is a very well written, clear manuscript describing a quite complex topic and dataset. It is important for those working in rotavirus and enteric vaccines in general, providing interesting results about the potential use of serological markers as correlates of protection against shedding following a 'challenge' dose of oral rotavirus vaccine. The results and discussion clearly present and discuss the implications of the findings. I have only very minor comments:

1. please provide a few more details on how the GMTs were adjusted at day 84 for maternal antibodies (why are they higher than unadjusted?)

2. a couple of typos ('two' on line 124 can be deleted; 'non-shedders' on line 319 should read 'shedders')

3. heterotypic neutralising antibody data are referred to on lines 188-191 but not presented

Author Response

This is a very well written, clear manuscript describing a quite complex topic and dataset. It is important for those working in rotavirus and enteric vaccines in general, providing interesting results about the potential use of serological markers as correlates of protection against shedding following a 'challenge' dose of oral rotavirus vaccine. The results and discussion clearly present and discuss the implications of the findings. I have only very minor comments:

  1. please provide a few more details on how the GMTs were adjusted at day 84 for maternal antibodies (why are they higher than unadjusted?)

Response: The following details have been added to the text in the methods section, with reference to the original study: “IgG and NAb titers at the one-month post third-dose time-point were adjusted for decay in maternal antibodies using the half-life calculated from participants in the placebo group who had detectable baseline titers that were higher than at the post-injection visit. This was established separately for each assay. Adjusted seroresponse was defined as a four-fold or more increase in titer from baseline to one month after receiving three doses of P2-VP8-P[8] vaccine or placebo (adjusted titer) in infants with an unadjusted post-injection ti-ter greater than the limit of detection [12].”

At baseline (pre-vaccination), most of the IgG and NABs would be maternally derived. At the time of the post-vaccination blood draw at ~18 weeks of age, the maternal antibody titres would have decreased, thus contributing a lower proportion of the total IgG/NAb titres. Adjustment for that decay would therefore lead to higher titres compared to the unadjusted titres. Maternal antibody titres would be a confounder in the comparison between baseline (pre-vaccination) and post-vaccination titres.

  1. a couple of typos ('two' on line 124 can be deleted; 'non-shedders' on line 319 should read 'shedders')

Response: Thank you for picking up these typos. They have been corrected.

  1. heterotypic neutralising antibody data are referred to on lines 188-191 but not presented

Response: The primary paper reported overall results of the heterotypic NAb data in vaccine compared to placebo recipients at the 18 week timepoint. Low neutralising antibody responses to heterologous P[4] and P[6] strains were demonstrated (Groome et al, 2017). Based on these results, a trivalent P2-VP8 vaccine was developed rather than further evaluation of the P2-VP8-P8 vaccine. Thus the authors felt it does not add much to the paper by presenting these data in detail. In light of this, the heterotypic results in lines 188-191 have been removed from the paper and a sentence was added to the methods as follows: “As low NAb responses to heterologous P[4] and P[6] strains were previously demonstrated [12], and responses against DS-1 and 1076 strains were not measured at the six-month post-vaccination time point, only NAb against Wa and 89-12 were assessed in the current analysis.”

Reviewer 2 Report

The manuscript is well written and presents very good data in the field especially for the evaluation of candidate parenteral vaccines.

The manuscript is well written

Author Response

The manuscript is well written and presents very good data in the field especially for the evaluation of candidate parenteral vaccines.

No comments that require a response from the authors.

Reviewer 3 Report

The Manuscript entitled “Association between immunogenicity of a monovalent parenteral P2-VP8 subunit rotavirus vaccine and fecal shedding of rotavirus following Rotarix challenge during a randomized, double-blind, placebo-controlled trial” analyzed the virus shedding and Ab responses after the administration of a parenteral subunit vaccine followed by a dose of a live attenuated vaccine (Rotarix) in children. It is important to remark that the virus shedding to be detected in the vaccine virus, however, the author did not show the typing of the RV positive samples, so natural infections, might be happing and the same time of vaccination and this confounding factor was not clarified in the study.

Material and Methods

First of all, I suggest the authors to add a Figure explaining the experimental design with all the different doses of vaccine tested, the point of challenge with Rotarix and the sampling points. Also strongly clarify that your are looking for a vaccine shedding.

Results

Section 3.1. This section could  have a graph showing the statistical differences detected in virus shedding among groups.

Section 3.2 

Table 1, this table maybe can also be replaced by a graph adding the results of the placebo group, because in its present form does not show any significant difference between vaccine shedder and no shedder within the vaccine group and the table can be transferred to supl material.  If author still decide to retain the tables, the significant different could be highlighted in bold.

Table 2, you can replace this table with a graph panel of the Ab responses (curves) through time and show with an asterisk the point where you detected significant differences between shedder and no-shedders. In this case, the statistical analysis should be a general mix model of repeated measures through time, using a AR1 variance covariance matrix or similar and including the child as a variable factor. How did the authors do this analysis? 

It is not clear how the adjustment by passive maternal antibodies was done? Your estimated that the responses are higher than the IgG titers actually detected, but protection is related to the existing antibodies. Please explain these results further.

Section 3.2 Placebo Table 3, same suggestion than previous Tables. 

Why the author did not compare between placebo and vaccinated group inside shedders and no-shedders?

In the discussion section, it is important to remember that this shedding is due to an attenuated strain not a natural infection with a virulent virus, so it is hard to correlate the obtained results regarding vaccine virus shedding and the correlation with protection to the disease.

 Line 334 It is well known by all rota virologists that  VN Ab titers are G-type and P-type specific. And that 80% of the VN activity is due to Ab against VP4 and not VP7.

Line 341.342. …This suggests that RV shedding after ORV challenge may be reflective of vaccine take following ORV and may be predictive of higher serum immune responses at this later time point. >>> Of course! This has been well demonstrated in animal models and humans.

Lines 355-361. All the limitations of the study highlighted by the author are very important and really represent a group of confounding factors that limit the conclusion that can be taken from the present analysis. I really suggest the author to get in contact with groups of statisticians specialized in this type of data, so they can improve the analysis and the way to show the results.

Serum Ab responses does not always correlate with the Ab present in the gut. Mainly after parenteral vaccinations. Probably the measure of IgG and IgA in the fecal samples (copro Ab responses) in the shedder and not shedder must give better association with the shedding of the vaccine virus.

Author Response

The Manuscript entitled “Association between immunogenicity of a monovalent parenteral P2-VP8 subunit rotavirus vaccine and fecal shedding of 3 rotavirus following Rotarix challenge during a randomized, 4 double-blind, placebo-controlled trial” analyzed the virus shedding and Ab responses after the administration of a parenteral subunit vaccine followed by a dose of a live attenuated vaccine (Rotarix) in children. It is important to remark that the virus shedding to be detected in the vaccine virus, however, the author did not show the typing of the RV positive samples, so natural infections, might be happing and the same time of vaccination and this confounding factor was not clarified in the study.

Response: “The strain was confirmed as the Rotarix® vaccine strain in 29/32 (91%) of shedders, with the other three (one vaccine recipient and two placebo recipients) being a G9P[8] strain, which was predominant in the 2015 rotavirus season in South Africa, indicating that they were exposed to a natural infection [12]. As we were assessing the use of ELISA to detect shedding in this study, we did not exclude these participants from the analysis. The majority were shedding the vaccine strain, so would have had a minimal effect on the results”. The authors have included this text as a limitation in the discussion.

Material and Methods

First of all, I suggest the authors to add a Figure explaining the experimental design with all the different doses of vaccine tested, the point of challenge with Rotarix and the sampling points. Also strongly clarify that your are looking for a vaccine shedding.

Response: A figure has been included in the supplementary material to show the time points of the study procedures. We have referred to “rotavirus shedding” in the paper rather than “vaccine shedding” as we included 3 samples which were rotavirus positive on Elisa, yet not with a vaccine strain. As we were assessing the use of ELISA to detect shedding in this study, we did not exclude these participants from the analysis.

Results

Section 3.1. This section could have a graph showing the statistical differences detected in virus shedding among groups.

Response: As this comparison could very simply be stated in the text, the authors did not feel that this warranted a figure.

Section 3.2

Table 1, this table maybe can also be replaced by a graph adding the results of the placebo group, because in its present form does not show any significant difference between vaccine shedder and no shedder within the vaccine group and the table can be transferred to supl material.  If author still decide to retain the tables, the significant different could be highlighted in bold.

Response: The main objective of this study was to assess the association between fecal rotavirus shedding and serum immune responses, so we would prefer to retain this table in the main body rather than the supplementary material. There were no significant differences in this table to highlight in bold.

Table 2, you can replace this table with a graph panel of the Ab responses (curves) through time and show with an asterisk the point where you detected significant differences between shedder and no-shedders. In this case, the statistical analysis should be a general mix model of repeated measures through time, using a AR1 variance covariance matrix or similar and including the child as a variable factor. How did the authors do this analysis?

Response: As shedding was measured at only one point in time, the authors felt that this would not be an applicable analysis technique and therefore we have used other methods suitable for time-invariant independent variables. These are described in the methods section. In order to use repeated measurement models with autoregressive covariance structures, the measurements of both the independent and dependent variables are expected to differ over time.

It is not clear how the adjustment by passive maternal antibodies was done? Your estimated that the responses are higher than the IgG titers actually detected, but protection is related to the existing antibodies. Please explain these results further.

Response: The following details have been added to the text in the methods section, with reference to the original study: “IgG and NAb titers at the one-month post third-dose time-point were adjusted for decay in maternal antibodies using the half-life calculated from participants in the placebo group who had detectable baseline titers that were higher than at the post-injection visit. This was established separately for each assay. Adjusted seroresponse was defined as a four-fold or more increase in titer from baseline to one month after receiving three doses of P2-VP8-P[8] vaccine or placebo (adjusted titer) in infants with an unadjusted post-injection ti-ter greater than the limit of detection [12].”

At baseline (pre-vaccination), most of the IgG and NABs would be maternally derived. At the time of the post-vaccination blood draw at ~18 weeks of age, the maternal antibody titres would have decreased, thus contributing a lower proportion of the total IgG/NAb titres. Adjustment for that decay would therefore lead to higher titres compared to the unadjusted titres. Maternal antibody titres would be a confounder in the comparison between baseline (pre-vaccination) and post-vaccination titres. In addition, as this was an immunogenicity study, we were unable to measure protection against rotavirus disease. We could only measure the fold increase from baseline (pre-vaccination) to post-vaccination, and maternal antibodies are a potential confounder in this comparison.

Section 3.2 Placebo Table 3, same suggestion than previous Tables.

Response: The authors have retained the table and highlighted significant differences in bold in the tables.

Why the author did not compare between placebo and vaccinated group inside shedders and no-shedders?

Response: The main objective for this analysis was to evaluate the association between fecal rotavirus shedding and immune responses, not to compare vaccination status within shedders and non-shedders.  Differences in shedding between vaccinees and placebo recipients were presented in the primary paper (Groome et al, 2017). For ease of interpretation of the results, we performed separate analyses for the vaccine and placebo groups.

In the discussion section, it is important to remember that this shedding is due to an attenuated strain not a natural infection with a virulent virus, so it is hard to correlate the obtained results regarding vaccine virus shedding and the correlation with protection to the disease.

Response: We acknowledge this limitation and have included this in the discussion as follows: “Rotavirus shedding was due to an attenuated stain, not natural infection with a virulent virus, making it difficult to correlate the findings with protection against disease.”

 Line 334 It is well known by all rota virologists that  VN Ab titers are G-type and P-type specific. And that 80% of the VN activity is due to Ab against VP4 and not VP7.

Response: The text has been changed to “The correlation between serum NAb response to different RV strains and protection against RV infection is context and strain specific.”

Line 341.342. …This suggests that RV shedding after ORV challenge may be reflective of vaccine take following ORV and may be predictive of higher serum immune responses at this later time point. >>> Of course! This has been well demonstrated in animal models and humans.

Response: The text has been changed to: “This shows that RV shedding after ORV challenge is reflective of vaccine take following ORV and is predictive of higher serum immune responses at this later time point.”

Lines 355-361. All the limitations of the study highlighted by the author are very important and really represent a group of confounding factors that limit the conclusion that can be taken from the present analysis. I really suggest the author to get in contact with groups of statisticians specialized in this type of data, so they can improve the analysis and the way to show the results.

Response: This is a secondary analysis of data obtained from the primary clinical trial to evaluate safety and immunogenicity of the P2-VP8-P8 vaccine. While there are limitations as described, the current analysis adds to existing literature. The authors view the analysis appropriate for the question they are trying to address.

Serum Ab responses does not always correlate with the Ab present in the gut. Mainly after parenteral vaccinations. Probably the measure of IgG and IgA in the fecal samples (copro Ab responses) in the shedder and not shedder must give better association with the shedding of the vaccine virus.

Response: We agree with your comment and will consider this for further evaluation of the samples and future studies. The following has been added to the discussion: “Rotavirus-specific copro-IgA and IgG antibodies were not measured in this study so the association with fecal RV shedding could not be assessed.”

Reviewer 4 Report

In this study, the authors aimed to identify the relationship between protection against Rotarix vaccine (challenge) virus shedding and the P2-VP8-P[8] vaccine-induced immune responses. Serum IgA and IgG titers directed against P2-VP8-P[8] vaccine antigen (anti-P[8] IgA and IgG) and the whole rotavirus lysate (anti-RV IgA) were measured, as well as neutralizing antibody titers against P[8], P[6] and P[4] RV strains in total 135 infants. No significant associations were identified between serum RV-specific immune responses measured one-month post-P2-VP8 vaccination and fecal RV shedding post-Rotarix challenge.

Although no correlate of protection was identified in this study, the finding that those infants shedding RV after the first Rotarix dose demonstrated a robust anti-RV IgA response later is very important. It indeed supports the use of shedding following 1st dose of oral RV vaccine as a predictor of vaccine take.  There will be added benefits of using a noninvasive sampling, such as measuring nasal Rotarix virus shedding. It is important to publish these results and to discuss the implications of the findings.

Line 129. I suggest revising subtitle 3.1 to “Association between vaccination status and non-immunological factors and fecal shedding of rotavirus”. The observed significantly lower shedding in vaccine than placebo recipients is an important result and should be reflected in the subtitle.

Author Response

In this study, the authors aimed to identify the relationship between protection against Rotarix vaccine (challenge) virus shedding and the P2-VP8-P[8] vaccine-induced immune responses. Serum IgA and IgG titers directed against P2-VP8-P[8] vaccine antigen (anti-P[8] IgA and IgG) and the whole rotavirus lysate (anti-RV IgA) were measured, as well as neutralizing antibody titers against P[8], P[6] and P[4] RV strains in total 135 infants. No significant associations were identified between serum RV-specific immune responses measured one-month post-P2-VP8 vaccination and fecal RV shedding post-Rotarix challenge.

Although no correlate of protection was identified in this study, the finding that those infants shedding RV after the first Rotarix dose demonstrated a robust anti-RV IgA response later is very important. It indeed supports the use of shedding following 1st dose of oral RV vaccine as a predictor of vaccine take.  There will be added benefits of using a noninvasive sampling, such as measuring nasal Rotarix virus shedding. It is important to publish these results and to discuss the implications of the findings.

Line 129. I suggest revising subtitle 3.1 to “Association between vaccination status and non-immunological factors and fecal shedding of rotavirus”. The observed significantly lower shedding in vaccine than placebo recipients is an important result and should be reflected in the subtitle.

Response: The subtitle has been revised as suggested.

Round 2

Reviewer 3 Report

The authors answered most of the queries.

The MS is acceptable for publication in its present form